Experimental evidence of chemical attraction in the mutualistic zebra mussel-killer shrimp system

Rolla Matteo 1
Consuegra Sofia 1
Carrington Eleanor 1
Hall David J. 2
Garcia de Leaniz Carlos c.garciadeleaniz@swansea.ac.uk 1
1 Department of BioSciences, Centre for Sustainable Aquatic Research, Swansea University , Swansea , United Kingdom
2 Cardiff Harbour Authority , Cardiff , United Kingdom
Garant Dany
Electronic publication date: 2019 Nov 21
Publication date: 2019
Volume: 7
Electronic Location ID: e8075
Received 2019 May 7; Accepted 2019 Oct 21
Copyright: ©2019 Rolla et al.
Copyright year: 2019
Copyright holder: Rolla et al.
License: This is an open access article distributed under the terms of the Creative Commons Attribution License, which permits unrestricted use, distribution, reproduction and adaptation in any medium and for any purpose provided that it is properly attributed. For attribution, the original author(s), title, publication source (PeerJ) and either DOI or URL of the article must be cited.
License URL: https://creativecommons.org/licenses/by/4.0/

Keywords: Synergy, Invasive species, Zebra mussel, Killer shrimp, Kairomones, Invasion meltdown, Sympatry, Group behaviour, Chemical attraction, Invasion facilitation

Funding: European Commission Horizon 2020 Aquainvad-ED project ITN-2014-ETN-642197 Funding was provided by the European Commission Horizon 2020 Aquainvad-ED project (Marie Sklodowska-Curie ITN-2014-ETN-642197) led by Sofia Consuegra. The funders had no role in study design, data collection and analysis, decision to publish, or preparation of the manuscript.

==============================
Invasion facilitation, whereby one species has a positive effect on the establishment of another species, could help explain the rapid colonisation shown by some freshwater invasive species, but the underlying mechanisms remain unclear. We employed two-choice test arenas to test whether the presence of zebra mussel (Dreissena polymorpha) could facilitate the establishment of the killer shrimp (Dikerogammarus villosus). Killer shrimp preferred to settle on mats of zebra mussel, but this was unrelated to mat size, and was not different from attraction shown to artificial grass, suggesting that zebra mussel primarily provides substrate and refuge to the killer shrimp. Killer shrimp were strongly attracted to water scented by zebra mussel, but not to water scented by fish. Chemical attraction to the zebra mussel’s scent did not differ between sympatric and allopatric populations of killer shrimp, suggesting that chemical attraction is not an acquired or learned trait. Our study shows, for the first time, chemical attraction between two highly invasive freshwater species, thereby providing a plausible mechanism for invasion facilitation. This has implications for managing the spread of killer shrimp, and perhaps other freshwater invasive species, because chemical attraction could significantly increase establishment success in mutualistic systems. Failure to consider invasion facilitation may underestimate the risk of establishment, and likely also the impact of some aquatic invaders.

Introduction

The impact of biological invasions has often been examined in isolation, under the implicit assumption that invaders do not interact with each other (Consuegra et al., 2011; Vanhaecke et al., 2015; Young et al., 2010; Young et al., 2009). However, invasion facilitation, whereby one species has a positive effect on the introduction, establishment or dispersal of other invasive species, is well documented, particularly in terrestrial plants and animals (Adams, Pearl & Bruce Bury, 2003; Altieri et al., 2010). For example, the presence of the European honey bee (Apis mellifera) has increased the reproduction success of the invasive shrub Lantana camara in Australia (Goulson & Derwent, 2004), and similar positive synergies among invasive species have also been reported across many taxa (Woodward et al., 1990). This led Simberloff & Von Holle (1999) to coin the term ‘invasional meltdown’ to describe the process by which the negative impacts triggered by one invasive species could be exacerbated by the interactions with other exotic species (Simberloff, 2006).

However, invasion facilitation has not received as much attention in freshwater habitats as it has in terrestrial ecosystems, possibly because it is more difficult to detect (Ricciardi, 2005), and because it typically only benefits one species (i.e., commensalism, Laihonen & Furman, 1986; Ricciardi, 2005). This lack of information is unfortunate because freshwater habitats, particularly lakes and ponds, rank among the most threatened ecosystems in the world, and this makes them particularly vulnerable to the threat of aquatic invasive species (AIS; Dudgeon et al., 2006). Habitat suitability models for invasive species, as well as risk maps (Crall et al., 2013; Jiménez-Valverde et al., 2011), rarely take into account the fact that some invaders can bioengineer their new habitat to suit their needs, or that the presence of one invasive species may make the habitat more attractive to other invaders (Strayer, 2012). Thus, the synergistic effects of invasive species and their cumulative impacts on native fauna may be underestimated in aquatic habitats if invasion facilitation exists and is not taken into account.

Two aquatic invaders that often occur together and may benefit from invasion facilitation are the zebra mussel (Dreissena polymorpha) and the killer shrimp (Dikerogammarus villosus). The two species are included in the 100 worst invasive species in Europe (http://www.europe-aliens.org), and in the case of zebra mussel, in the world (Lowe et al., 2000). Zebra mussels form dense mats on natural and artificial substrates which require expensive eradication programmes (Lovell, Stone & Fernandez, 2006), and compete directly for food and space with native bivalves (Fahnenstiel et al., 1995a; Fahnenstiel et al., 1995b; Johengen et al., 1995), sometimes driving them to extinction (Baker & Hornbach, 1997). The killer shrimp, on the other hand, has the typical profile of an efficient and plastic invader (Rewicz et al., 2014; Rossano, Di Cristina & Scapini, 2013), being able to adapt to a wide range of waters and conditions (Piscart et al., 2003). Its wide diet includes many macroinvertebrates, including native gammarids, which may be displaced and driven to local extinction (Dick & Platvoet, 2000; Piscart et al., 2003).

Both species share a broad, common Ponto-Caspian geographical origin, but the extent of sympatry in their native, as well as in the invaded areas, is unclear. The zebra mussel was first reported in Europe during the nineteenth century, becoming highly invasive and forming large populations (Son, 2007). In contrast, the killer shrimp is a much more recent invader, it has only been detected in Europe during the last 25 years, but has spread rapidly ever since (MacNeil et al., 2010; Rewicz et al., 2014; Rewicz et al., 2017; Tricarico et al., 2010).

While many of the sites colonised by the killer shrimp in Europe had already established populations of zebra mussel, that might be indicative of invasion facilitation (Gallardo & Aldridge, 2015), the killer shrimp has also invaded many areas devoid of zebra mussel (Rewicz et al., 2015; Van der Velde, Rajagopal & Bij de Vaate, 2010). Thus, whether there have been synergies in the establishment of these two species is not clear (Devin et al., 2003). The zebra mussel has shown mutualistic interactions with one gastropod (Ricciardi, 2005; Ricciardi, Whoriskey & Rasmussen, 1997), and two macrophytes (MacIsaac, 1996; Skubinna, Coon & Batterson, 1995), and can benefit the killer shrimp in various ways. For example, the dense interstitial matrix formed by the shells of zebra mussels may provide refuge for the killer shrimp (Ricciardi, Whoriskey & Rasmussen, 1997), allowing it to survive outside the water, while the production of faeces and pseudo faeces may provide food (Gergs & Rothhaupt, 2008a; Gergs & Rothhaupt, 2008b; Ricciardi, Whoriskey & Rasmussen, 1997; Stewart, Miner & Lowe, 1998b). Similarly, zebra mussel larvae can attach to the hard chitin cover of the killer shrimp which could facilitate their dispersal (Kenderov, 2017; Yohannes et al., 2017).

Given their common geographical origin, and recent evidence suggesting that the two species often occur together and might benefit each other, we hypothesized that killer shrimp might be chemically attracted to the presence of zebra mussel. We further hypothesized that attraction might differ depending on whether killer shrimp were found in habitats already colonised by the zebra mussel, i.e., whether attraction differed between sympatric and allopatric conditions.

Previous studies had indicated that mussel beds may provide killer shrimp with food and shelter (Gergs & Rothhaupt, 2008a; Gergs & Rothhaupt, 2008b), but two-choice preference tests yielded contradictory results (Gergs & Rothhaupt, 2008a), and did not consider the effects of coexistence or group dynamics on killer shrimp behaviour (Truhlar & Aldridge, 2015). We therefore employed an experimental approach to examine the attraction of killer shrimp to zebra mussel, considering potential differences between sympatric and allopatric populations, and individual versus group behaviour. In the first experiment we tested if killer shrimp had a preference for high densities of zebra mussels, as one might expect from a mutualistic system. In the second experiment, we tested if killer shrimp had a preference for live zebra mussels compared to empty shells or an artificial substrate, and whether this was affected by group behaviour, as one might expect if attraction was mostly driven by food, and not by cover; in the third experiment we tested if killer shrimp could detect the presence of zebra mussel through chemical cues in the water, and whether this depended on past coexistence. Ultimately, our aim was to address some of the underlying mechanisms of invasion facilitation as this might help design better predictive models and more effective control measures of these two aquatic invaders.

Materials and Methods

Collection and origin of samples

Sympatric zebra mussel and killer shrimp were collected from Cardiff Bay (Cardiff, UK—Grid reference: ST 19210 73510) in April 2016, whereas allopatric killer shrimp were collected from the Upper Mother Ditch (Margam, UK, Grid reference: SS 79029 85506) in September 2016, where the zebra mussel is not yet present (Fig. 1). These were brought to the CSAR facilities at Swansea University and maintained in 6 × 20 L tanks fed by separated recirculation aquaculture systems, with a weekly replacement of c. 20% volume. Zebra mussels (∼2.4 kg) were fed three times per week with a 2 L mixture of Scenedesmus sp. and Chlorella sp., while killer shrimps (∼100 g) were fed three times per week with 6 g of frozen bloodworms. Water temperature was maintained at 15–16.5 °C. The killer shrimp used in the tests had an average size of 16.8 ± 0.9 mm and were tested at water temperature ranging between 15.4 °C and 16.2 °C.

Figure 1 Location of experimental killer shrimp populations living in sympatry (Cardiff Bay) and allopatry (Upper Mother Ditch) with zebra mussel.

Experiment 1. Preference by killer shrimp of zebra mussel density

To test if killer shrimp had a preferred density of zebra mussel to settle on, we employed a 3L tank (L25 × H 10 × W12 cm; Fig. 2A) divided into two equal sections, each with a different density of zebra mussel (0, 33, 67, or 100% cover), and an acclimatisation plastic cylinder in the middle. Individual killer shrimp (n = 96) were allowed to acclimatise for five minutes in the cylinder, then the cylinder was lifted and the position and behaviour (swimming or hiding) of the shrimp after 20 min was recorded. We assumed that if the shrimp was hiding it meant it had found a suitable substrate, whereas if it was still swimming it meant it was still looking for a refuge. We tested the killer shrimp’s binary choice over six matched densities (n = 16) of zebra mussel: 0–33%, 0–66%, 0–100%, 33–66%, 33–100%, and 66–100% employing a total of 96 specimens, and allocating the densities to the left or right sides of the test arena at random.

Figure 2 Experimental set up used to test substrate preferences of killer shrimp (A, side view; C, top view) and chemical attraction to water scented by zebra mussel (B, side view; D, top view).

Experiment 2. Preference for zebra mussel over an artificial substrate

To test if the attraction of killer shrimp for zebra mussel-beds was simply related to the presence of cover or to other factors (such as bio-deposited material) we compared preference for living shells against either empty shells of zebra mussel or artificial grass (PE thickness 15 mm) of similar texture and extent of refuge. We used one killer shrimp per trial (n = 30), and then twenty killer shrimp per trial (n = 80) to understand if substrate choice was affected by group dynamics. The experimental protocol was the same as in Experiment 1, but in this case each side of the test arena afforded 50% cover and we used a 20 L test tank (L40 × H15 × W35 cm).

Experiment 3. Chemical attraction to zebra mussel

To test if killer shrimp was chemically attracted to the scent of zebra mussel we employed a simplified version of the two-choice Perspex fluviarium used by Kroon (2005) to test the preference of another crustacean (Fig. 2B). The fluviarium consisted of an acclimatization chamber (L5 × H6.5 × W5 cm) and two 0.3L choice chambers (L20 × H6.5 × W10 cm) with a total volume of approximately 0.7L. We tested preferences against dechlorinated tap water (blank), water scented with zebra mussel and Nile tilapia (Oreochromis niloticus) to control for possible attraction to organic matter, as well as blank water vs blank water to control for chamber bias. Individual killer shrimp were allowed to acclimatize for five minutes, the valves connected to the two water inlets were opened, the gate was lifted, and the time spent in each chamber was recorded for 15 min with a GoPro Hero camera. We compared the time spent in each arm as well as the number of transitions between arms as a measure of activity. Scent drip dosage was adjusted at 200 ml/min.

To prepare the scented water we placed either zebra mussels or fish (tilapia) in a tank filled with dechlorinated water for 24 h at a biomass of 50 g/L. The fluviarium was drained, cleaned with 90% ethanol and rinsed with fresh water between trials to remove potential chemical cues that could affect the next experiment. We repeated the experiment with killer shrimp originating from a population living in sympatry (Cardiff Bay, n = 60) or allopatry (Upper Mother Ditch, n = 60) with zebra mussels. All the zebra mussel came from Cardiff Bay.

Statistical analysis

We used R 3.3 (R Core Team, 2017) for all analysis. In experiments 1 and 2, we used a generalized linear model (GLM) with a binomial log-link to test if the number of killer shrimp in the scented arm differed with treatment, and we then used a two-sided binomial test to assess if there was a statistically significant preference for the high or low density (Experiment 1) or different substrate combinations (Experiment 2) at each binary choice comparison. For Experiment 3 (two-way choice fluviarium), we used a linear model with time spent in the scented arm as the dependent variable and origin (allopatry vs sympatry) and type of scent (blank, zebra mussel, tilapia) as the predictors; we then used paired t-tests to assess which type of matched scent comparisons was statistically significant.

Ethics statement

Zebra mussels and killer shrimp were collected under sampling permit CHA-01042016 from the Cardiff Water Authority. All experiments were carried out in accordance with Swansea University Ethical guidelines and were approved by the College of Science Ethics Committee (300419/1557). Water removed from the experimental system was treated with bleach before disposing, to avoid accidental dispersion of zebra mussel larvae.

Results

Experiment 1. Preferred zebra mussel densities

Preference for zebra mussel varied depending on the densities being compared (χ2 = 29.09, df = 1, P < 0.001). Killer shrimp showed a clear preference for the side of the tank with zebra mussel when the alternative was a bare tank bottom (Figs. 3A–3C; binomial proportion test: 0–33% P = 0.004; 0–66% P < 0.001; 0–100% P < 0.001). However, when both sides of the test arena had different densities of zebra mussel, killer shrimp showed no preference (Figs. 3D and 3F; binomial proportion test 33–66%, P = 0.454; 66–100%, P = 1.00) or preferred the lower density (Fig. 3E; 33–100%, P = 0.004). After 20 min, the majority of killer shrimp (85/96 or 88.5%) were found to be hiding, rather than swimming (binomial proportion test P < 0.001) regardless of treatment (χ2 = 1.745, df = 1, P = 0.883).

Figure 3 Proportion of individual killer shrimp (binomial 95 CI) settling in zebra mussel beds of different sizes in six binary choice tests (A–F, n = 16 shrimp/test) involving different amount of zebra mussel cover (0, 33, 66, and 100% tank cover).

Experiment 2. Substrate preference

When tested individually, killer shrimp did not prefer live zebra mussels over artificial grass (Fig. 4A; binomial proportion test, P = 0.584), or over empty zebra mussel shells (binomial proportion test, P = 0.200; Fig. 4B). The majority of individuals were found hiding (rather than swimming), both when the comparison was against artificial grass (binomial proportion test, 83.3% P < 0.001) and also when there were empty shells (binomial proportion test, 76.6% P = 0.005).

Figure 4 Proportion of killer shrimp (binomial 95 CI) settling in binary choice tests involving different substrates (live zebra mussel, empty shells of zebra mussel, artificial grass) tested singly (A–B, n = 30 shrimp/test) or in groups of 20 (C–D, n = 4 tests).

However, when the experiment was repeated with 20 shrimp per trial (4 trials or 80 shrimp), killer shrimp strongly preferred the zebra mussel substrate over artificial grass (Fig. 4C; binomial proportion test, P = 0.006) and also over empty shells (Fig. 4D; binomial proportion test, P = 0.006). As before, at the end the trials the majority of individuals were hiding, both when the comparison was against artificial grass (binomial proportion test, 77.5% P < 0.001) and also against empty shells (binomial proportion test, 72.5% P = 0.006).

Experiment 3. Chemical attraction to zebra mussel

No side preference was detected when killer shrimp were tested against blank water in both arms of the 2-choice fluviarium, either in the sympatric (t9 = 1.343, P = 0.212; Fig. 5A) or allopatric killer shrimp populations (t19 =  − 1.280, P = 0.216; Fig. 5B), indicating that there was no side bias. When killer shrimp were tested against water conditioned with tilapia scent, no preference was observed over blank water, either in sympatry (t19 = 0.819, P = 0.423; Fig. 5C) or allopatry (t19 =  − 0.687, P = 0.500; Fig. 5D). However, when killer shrimp were tested against water conditioned with zebra mussel scent, there was a strong chemical attraction to the zebra mussel scent, both in the sympatric (t27 =  − 2.176, P = 0.038; Fig. 5E) and allopatric population (t19 =  − 2.614, P = 0.017; Fig. 5F). Chemical attraction for zebra mussel scent was equally strong in the sympatric and allopatric populations (F1,118 = 1.036, P = 0.311).

Figure 5 Preference by individual killer shrimp (mean time spent, s ± 95 CI) in water conditioned with different scents (blank water, n = 40; tilapia scent, n = 40; zebra mussel scent, n = 40) from sympatric (A, C, E; n = 60) and allopatric (B, D, F; n = 60) populations.

The analysis of activity (measured as the number of transitions between arms) indicates that activity was influenced by the type of test scent (Fig. 6), as killer shrimp made more changes when both arms were dosed with blank water than when one arm was dosed with zebra mussel scent (P = 0.002) or tilapia scent (P < 0.001). No difference in activity was observed when killer shrimp were presented with the scent of zebra mussel or the tilapia scent against blank water (P = 0.759). Overall, the allopatric population made more choices and was more active than the sympatric population (P = 0.005).

Figure 6 Activity (mean number of transitions ± 95 CI) of individual killer shrimp tested in water conditioned with different scents (blank water, n = 40; tilapia scent, n = 40; zebra mussel scent, n = 40) from sympatric (A; n = 60) and allopatric (B; n = 60) populations.

Discussion

Our study provides novel experimental insights into some of the potential underlying reasons for the joint occurrence of zebra mussel and the killer shrimp, two of the world’s worst aquatic invaders (Lowe et al., 2000). We found that killer shrimp showed a strong tendency for hiding in zebra mussel beds, and were also chemically attracted to the scent of zebra mussels, which may facilitate their invasion.

In our experiments, killer shrimp consistently avoided the empty side of the tank (substrate coverage 0%) that did not afford any refuge, and generally preferred to settle on zebra mussel beds, even when tested with blank water and without any threat of predation. The strong preference for a substrate that offers refuge is in agreement with observations under natural conditions, where the species is typically found living among gravel, cobbles and boulders, and absent in places where there is silt or substrates that do not afford refuge (Boets et al., 2010; MacNeil et al., 2010). While juvenile killer shrimp may also be found living among macrophytes (Devin et al., 2003), as happens for juveniles of several other predatory amphipods (Berezina, 2007), adults tend to prefer hard substrates of large grain size, including cobble and roots (Devin et al., 2003), pebbles (Van Riel, Van der Velde & Bij de Vaate, 2009), fissured stones (Kley et al., 2009), and coarse gravel (Boets et al., 2010). Kobak, Jermacz & Dzierzyńska-Bialończyk (2015) have suggested that the substrate preference of the killer shrimp is size dependent and determined by the interstitial spaces of the substrate, as this influences ease of movement and the ability to find refuge. The species appears to choose fissures that closely match its body size (Platvoet et al., 2009), which might explain why they prefer larger substrates as they become older. In this sense, zebra mussel beds provide an ideal refuge for juveniles and adults alike, because as the mussels grow the interstitial spaces also become larger.

The findings of Experiment 2, where we tested the preference of killer shrimp for zebra mussel over other textured substrates, are more difficult to interpret as different results were obtained depending on group size. When killer shrimp were tested singly, no preference was detected for live zebra mussels over empty shells or artificial grass of similar texture, suggesting that substrate preference was mainly governed by the availability of refuge, which previous experiments have shown confers protection from fish predators (Kinzler & Maier, 2006). However, when groups of twenty shrimp were tested, a strong preference for live zebra mussel over other substrates was found, suggesting the existence of group behaviour that cannot solely be explained by refuge availability and deserves further investigation. Killer shrimp tend to form aggregations, and these are thought to be advantageous and increase the chances of successfully colonising new areas (Truhlar & Aldridge, 2015). However, while group behaviour can increase fitness and reproductive success (Réale et al., 2007), it can also facilitate intraspecific predation (cannibalism), which is frequently observed in amphipods (Dick, Montgomery & Elwood, 1993; Hunte & Myers, 1984; Ward, 1985), including the killer shrimp (Dick & Platvoet, 2000; Dick, Platvoet & Kelly, 2002; MacNeil, Dick & Elwood, 1997). An inverse association may exist between sociability and cannibalism in amphipods (Kinzler et al., 2009; Truhlar & Aldridge, 2015), although this may also be influenced by predation pressure (Dick, Montgomery & Elwood, 1993). Cannibalism in killer shrimp does not appear to be so strong as to reduce the species’ sociability, possibly because cannibalism mainly targets small juveniles (Kinzler & Maier, 2003) which tend to be spatially segregated from larger adults that could prey on them (Devin et al., 2003).

One novel finding of our study was the strong chemical attraction shown by killer shrimp to the scent of live zebra mussels (or something associated with them), a response not seen to blank water or the scent of non-predatory fish. Amphipods use chemical cues as their main form of communicating between conspecifics (Thiel, 2011), and also to recognize and avoid predators (Wooster, 1998), but chemical attraction in contexts other than conspecific recognition or prey-predator interactions has, to our knowledge, not been reported before. A previous two-choice study reported chemical avoidance of zebra mussel scent by killer shrimp, but this was tested using water from a lake known to contain many predatory fish (Gergs & Rothhaupt, 2008a).

Chemical detection in amphipods is mediated mainly via specific sensillae located on the antennae (reviewed by Hallberg & Skog, 2011) and is used in mate choice and species discrimination, reducing the chances of interspecific mating between similar species (Cothran et al., 2013; Dick & Elwood, 1990). Chemical cues are also used by females to recognize and defend their offspring against conspecifics (Mattson & Cedhagen, 1989), and some amphipods can also recognize alarm cues from damaged conspecifics and mount a strong freezing response as an anti-predatory strategy (Sehr & Gall, 2016). Killer shrimp have been reported to use chemical cues to recognize and avoid potential predators such as the spiny-cheek crayfish Orconectes limosus (Hesselschwerdt et al., 2009), the European bullhead Cottus gobio (Sornom et al., 2012) and the racer goby Babka gymnotrachelus (Jermacz et al., 2017). Our study shows that killer shrimp can also use chemical cues to find zebra mussels that provide not only cover and refuge (Ricciardi, Whoriskey & Rasmussen, 1997), but also food through the production of faeces and pseudo faeces (Gergs & Rothhaupt, 2008a; Gergs & Rothhaupt, 2008b; Ricciardi, Whoriskey & Rasmussen, 1997; Stewart, Miner & Lowe, 1998a).

A preference for settling in zebra mussel beds was only manifested in our study when killer shrimp were tested in groups, not when they were tested singly, which serves to highlight the need to take group dynamics into account in behavioural studies of social gammarids (Williams, Navins & Lewis, 2016). The preference for zebra mussel beds shown by the killer shrimp (at least when they are tested in groups), and the fact that they are strongly attracted to the zebra mussel scent, suggests that this could be an example of invasion facilitation, as seen in other studies. For example, positive synergies resulting in invasion facilitation have been reported for an invasive algae and an invasive bryozoan (Levin et al., 2002), as well as among invasive fish parasites (Hohenadler et al., 2018). Other well known examples of invasion facilitation, include the bullfrog-sunfish system, where the survival of the invasive bullfrog (Rana catesbeiana) was enhanced by the presence of the non-native bluegill sunfish (Lepomis macrochirus) because the latter preyed on native dragonfly which in turn preyed on bullfrog tadpoles (Adams, Pearl & Bruce Bury, 2003). Similarly, predation by an invasive crab on a large native clam resulted in the spread of a smaller invasive clam due to competitive release (Grosholz, 2005).

Our study indicates that chemical attraction by the killer shrimp to the zebra mussel scent was as strong under sympatric as it was under allopatric conditions, suggesting this is not a recently acquired or learned trait, but rather an older behavioural adaptation. However, the absence of population replication (it is very difficult to find populations of killer shrimp that do not coexist with zebra mussel) makes it difficult to draw firm conclusions and would warrant further studies.

Conclusion

In general, mutualist interactions are less well studied than competitive ones (Simberloff & Von Holle, 1999), and interactions between invasive species are less well known than those between invasive and native species (Gallardo & Aldridge, 2018). There is still limited knowledge on positive interactions among invasive species, despite the fact that this may hold the key for more effective control of new invasions. Given the strong preference for settling on zebra mussel mats, and the benefits that this entails (Gergs & Rothhaupt, 2008b; MacNeil, Platvoet & Dick, 2008)—including not just refuge, but also benthic organic matter that can be a source of food (Ricciardi, Whoriskey & Rasmussen, 1997; Stewart, Miner & Lowe, 1998b), chemical attraction may help understand synergies between these two invasive species. Chemical attraction may need to be taken into account in the study of patterns of distribution of these two invasive species, since the presence of zebra mussel could make the killer shrimp more likely to become established, and hence a more successful invader. For example, chemical attraction to zebra mussels could help the establishment success of killer shrimp by decreasing search time for food and shelter. The zebra mussel’s ability to settle on a wide range of substrate (Marsden & Lansky, 2000) and bioengineer its own environment (Mayer et al., 2001) can turn formerly unsuitable habitats into favourable locations for killer shrimp settlement, expanding the potential range of suitable environments. For example, the NBN Atlas (https://nbnatlas.org/), the UK’s largest biodiversity database, holds 3,182 records of zebra mussel but only 317 observations of killer shrimp and information on the presence of zebra mussel could help make more accurate predictions of the likely spread of killer shrimp. This is of particular concern when the species is a recent invader since there is typically insufficient information to predict areas at risk or to guide management (Morales, Fernández & Baca-González, 2017). In Great Britain, the killer shrimp was first detected in 2010 (Rewicz et al., 2014) and is only present in 8 locations, making it difficult to predict its future dispersal (Rodríguez-Rey et al., 2019). The zebra mussel, in contrast, was first detected in 1824 and is now established in 376 locations in England and Wales (Aldridge, Elliott & Moggridge, 2004; Rodríguez-Rey et al., 2019), potentially increasing the number of favourable locations for killer shrimp many fold. Yet, current invasive species prioritisation lists (Boets et al., 2014; Carboneras et al., 2018) and risk assessment guidelines (Roy et al., 2018) tend to view invasive species in isolation, making no allowance for invasion facilitation. Our study suggests that information on the presence of zebra mussel should be incorporated into risk maps and models of killer shrimp dispersal, because ignoring chemical attraction will likely underestimate the extent and consequences of invasion facilitation.

Supplemental Information

Supplemental Information 1 Experiment 1

Zebra mussel density choice in killer shrimp

Click here for additional data file.

Supplemental Information 2 Experiment 2

Substrate choice in killer shrimp

Click here for additional data file.

Supplemental Information 3 Experiment 3

Chemical attraction in killer shrimp

Click here for additional data file.

We are grateful to Teja Muha (Swansea University), Emma Keenan and Graham Rutt (Natural Resources Wales) for help with the sampling and to Josh Jones (Swansea University) for help with the maps.

Additional Information and Declarations

Competing Interests

Author Contributions

Field Study Permissions

Data Availability

The authors declare there are no competing interests.

Matteo Rolla conceived and designed the experiments, performed the experiments, analyzed the data, prepared figures and/or tables, authored or reviewed drafts of the paper, approved the final draft.

Sofia Consuegra conceived and designed the experiments, contributed reagents/materials/analysis tools, authored or reviewed drafts of the paper, approved the final draft.

Eleanor Carrington performed the experiments, approved the final draft.

David J. Hall authored or reviewed drafts of the paper, approved the final draft, logistics, permits and collection of samples.

Carlos Garcia de Leaniz conceived and designed the experiments, analyzed the data, contributed reagents/materials/analysis tools, prepared figures and/or tables, authored or reviewed drafts of the paper, approved the final draft.

The following information was supplied relating to field study approvals (i.e., approving body and any reference numbers):

Permission to collect killer shrimp and zebra mussel was granted by The Cardiff Harbour Authority, permit CHA-01042016.

The Swansea University, College of Science Ethics Committee approved this research (300419/1557).

The following information was supplied regarding data availability:

Data are available at FigShare: Rolla, Matteo; Consuegra, Sofia; Hall, David; Carrington, Ellie; Garcia de Leaniz, Carlos (2019): Killer shrimp preferences. figshare. Dataset. https://doi.org/10.6084/m9.figshare.8080790.v1.

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
