# Peer review of "Experimental evidence of chemical attraction in the mutualistic zebra mussel-killer shrimp system"

_PeerJ, doi:10.7717/peerj.8075_

## Round 0.1 · original submission · Minor Revisions

We have now received two reviews for your manuscript and, as you will see, they were both positive. I am thus returning your manuscript for revisions that should deal with all comments raised by reviewers and in particular:

-Clearer justification for the experimental approach
-More thorough explanation of the study questions
-Streamlining of the discussion
-An improvement of figures and tables, including supplementary material.
-A correction of all grammar mistakes

Reviewer 1 ·

Basic reporting

BASIC REPORTING
Clear, unambiguous, professional English language used throughout
The language is clear; still there are a few grammar mistakes to revise (e.g. Line 189-missing subject) and long sentences that could be improved; Lines 222-223 should be rephrased: the difference in activity was observed in the shrimps tested with different scents. It is noted a redundancy in the use of hypothetical verbs (e.g. Lines; ). “Killer shrimp” has the plural “killer shrimps”, which should be used when referring to experimental specimens or populations, while the singular “Killer Shrimp” can be used to design the species.
Intro & background to show context.
The context is well shown; however, the need of an experimental approach is not wwll justified. A change of the title is suggested, with the title being “Chemical attraction” and “experimental evidence” a subtitle.
In the Abstract, at line 25, the statement “shows for the first time” should be supported by a bibliographic analysis, including several species, a review that would be beyond the objectives of the experimental study. At lines 24-25, the absence of "learning" was not shown by theexperiments - specific experiments could be designed for this, whereas, the chemical attraction observed appears a species-specific trait. At lines 27-28, “and perhaps other freshwater invasive species” is just a speculation, not supported by experimental data. Moreover, “perhaps” should be avoided in scientific papers. The keyword “synergy” is too general and could be applied in several other contexts.
Introduction - The definition of “plastic invader” (line 68) needs an explanation: invasive species may be physiologically or behaviourally plastic ( for behaviour, see: C. Rossano, G. Di Cristina, F. Scapini, 2013. Life cycle and behavioural traits of Dikerogammarus villosus (Sowinsky, 1894) (Amphipoda, Gammaridae) colonising an artificial fresh water basin in Tuscany (central Italy). CRUSTACEANA, 908-931).
Literature well referenced, relevant and updated.
Structure conforms to PeerJ standards and discipline norm.
Figures are relevant, high quality, well labelled & described.
Figure 1 should be integrated with a geographic map that locates Wales in a continental (European) context for international readers. This would also show the spread of the two invasive species, which is mentioned in the introduction and literature quoted. Figure 2 should be labelled to show where the shrimps were put into and then recorded, and the position of the substrates and chemicals they were subjected to. Figures 3-6 should report whether the killer shrimps were tested singly or in groups. In the figures where percentages are shown, the total number of tested individuals should be also reported.
Raw data supplied.
The day time when the experiments were conducted should be noted in the table. When percentages are shown, the total should also be given. For temperatures, the units should be added; in the excel table, please use the point for the decimals.

Experimental design

Original primary research within the Scope of the journal.
The paper reports a well conducted experimental study on chemical attraction of killer shrimps by zebra mussel.
Research question well defined, relevant & meaningful. It is stated how the research fills an identified knowledge gap.
However, the use of an experimental approach is not really justified, except by mentioning that it was done “for the first time”. The work hypotheses tested are just (with some shyness) put forward: “We hypothesized that killer shrimp might… further hypothesized that…might”. Moreover, there is a logical mistake in the use of “might” for a prediction. It is suggested to write a clear general hypothesis and list the two predictions.
Rigorous investigation performed to a high technical & ethical standard.
The experiments were well conducted and data correctly analysed. An ethics statement is included, important to avoid the dispersion of invasive species.
Methods described with detail & information to replicate.
It is suggested to add a table providing a synthetic description of the protocol used in the different experiments. Lines 113-117 of the Methods should be transferred to the Introduction, with the question and hypotheses to justify the approach, while the Methods should include the details of the experimental protocol. The heading of experiment 1 (Choice of zebra mussel density) is confusing because the subjects of the experiment were the killer shrimps, subjected to a choice. At line 147, the term “recognize” is not-properly used, because the choice was tested, not the "recognition", which would have implied an ad hoc experimental approach.

Validity of the findings

Impact and novelty are discussed with respect to management of biological invasions in freshwater bodies.
All underlying data have been provided; they are robust, statistically sound, & ontrolled.
The first statement in the Discussion at line 228 is speculative and was not shown by the results. It is suggested to stress the experimental evidence of the chemical attraction of the killer shrimp by the zebra mussel, while the reason (cause) for the joint occurrence of the two species may depend on several interacting factors. At lines 256 and successive, “group effect” is confounded with “social behaviour/sociability”, which was not shown by the experiments on killer shrimps: social behaviour would imply active interactions between individuals, while the group effect observed may have several different reasons (this could be discussed). Line 273: note that “chemical attraction” to other species is generally observed by predator species, including killer shrimps. The discussion at lines 274-287 extends to different amphipods, which should be listed, also mentioning their habitat (freshwater, bottom sea, rocky substrates, sandy substrates, intertidal, terrestrial, etc.), as amphipods are a large and diversified group. The result regarding the group facilitation reported at lines 288-289 should be better discussed, as it is relevant. At line 300-301, it is suggested that chemical attraction is an “older behavioural adaptation”, which, according to current evolutionary theory, is not contradictory to acquired behaviour – species-specific adaptation would be more appropriate to this context.
Conclusions are well stated, linked to original research question & limited to supporting results.

Additional comments

The manuscript is publishable, however it may be improved by:
1) Adding a justification for the experimental approach within the issue of invasive species control;
2) Avoiding subjective and hypothetical statements, which are not appropriate when reporting experimental evidence;
3) Improving the figures and tables, including the supplementary material, by locating the populations tested in an European dimension, adding the total when percentages are given and providing the daytime of the experiments (important in behavioural studies. A table with an overview of the experimental protocol would be useful;
4) Please, revise the English for grammar and syntax (e.g. use hypothetical verbs, long sentences)

·

Basic reporting

This paper was very well written.

A minor point is that scientific names are italicized in some references but not others.

Experimental design

The experimental design is fine, but the set up for the experiments is not adequate. As the Introduction ends, I appreciate the study questions but the questions do not fully address the experiments that were conducted. These should match better. For example, why look at density when it comes to attraction? What is the study question and why study it? Why look at group dynamics? Again, a good question but no set up for why it should be studied.

Validity of the findings

Findings are valid. I would appreciate more discussion about why as the authors say on lines 314-315 "the presence of zebra mussel could make the killer shrimp more likely to become established." This does not diminish the results, but I would like to see more speculation as to what they think is going on with this unique attraction.

---

## Round 0.2 · accepted · Accept

I checked the revised manuscript and the rebuttal letter and I am generally pleased with the modifications performed on the previous version.